# Comprehensive Response of *Rhodosporidium kratochvilovae* to Glucose Starvation: A Transcriptomics-Based Analysis

**DOI:** 10.3390/microorganisms11092168

**Published:** 2023-08-27

**Authors:** Meixia He, Rui Guo, Gongshui Chen, Chao Xiong, Xiaoxia Yang, Yunlin Wei, Yuan Chen, Jingwen Qiu, Qi Zhang

**Affiliations:** Faculty of Life Science and Technology, Kunming University of Science and Technology, Kunming 650500, China; MeixiaHe15@outlook.com (M.H.); guorui0731@foxmail.com (R.G.); cgs160627@outlook.com (G.C.); XiongChao0102@outlook.com (C.X.); XiaoxiaYang1212@outlook.com (X.Y.); homework19@126.com (Y.W.); cheny@kust.edu.cn (Y.C.); qiujingwenrr@outlook.com (J.Q.)

**Keywords:** *Rhodosporidium kratochvilovae*, glucose starvation, acetyl-CoA, carotenoids, reactive oxygen species, fatty acid β-oxidation, fatty acyl-CoA oxidase gene

## Abstract

Microorganisms adopt diverse mechanisms to adapt to fluctuations of nutrients. Glucose is the preferred carbon and energy source for yeast. Yeast cells have developed many strategies to protect themselves from the negative impact of glucose starvation. Studies have indicated a significant increase of carotenoids in red yeast under glucose starvation. However, their regulatory mechanism is still unclear. In this study, we investigated the regulatory mechanism of carotenoid biosynthesis in *Rhodosporidium kratochvilovae* YM25235 under glucose starvation. More intracellular reactive oxygen species (ROS) was produced when glucose was exhausted. Enzymatic and non-enzymatic (mainly carotenoids) antioxidant systems in YM25235 were induced to protect cells from ROS-related damage. Transcriptome analysis revealed massive gene expression rearrangement in YM25235 under glucose starvation, leading to alterations in alternative carbon metabolic pathways. Some potential pathways for acetyl-CoA and then carotenoid biosynthesis, including fatty acid β-oxidation, amino acid metabolism, and pyruvate metabolism, were significantly enriched in KEGG analysis. Overexpression of the fatty acyl-CoA oxidase gene (*RkACOX2*), the first key rate-limiting enzyme of peroxisomal fatty acid β-oxidation, demonstrated that fatty acid β-oxidation could increase the acetyl-CoA and carotenoid concentration in YM25235. These findings contribute to a better understanding of the overall response of red yeast to glucose starvation and the regulatory mechanisms governing carotenoid biosynthesis under glucose starvation.

## 1. Introduction

In natural environments, microorganisms can perceive fluctuations in nutritional conditions and respond appropriately through multiple sensing mechanisms, which allow cells to adapt to environments and survive [1]. For continuous growth, yeast requires various nutrients, such as carbon, nitrogen, phosphate, sulfate, metal ions, and vitamins [2]. Carbon sources are the basis for the growth, metabolism, and activity of cells [3]. Although yeast cells can utilize multiple carbon sources [4,5], they primarily prefer glucose, and after its exhaustion, they start using other carbon sources [6]. Yeast cells can sense the extracellular and intracellular glucose concentrations through complex mechanisms and change the fermentation mode to regulate cellular activities [3]. When glucose is available, *Saccharomyces cerevisiae* mainly functions in the fermentation mode, producing ethanol and acetic acid through glycolysis and ethanol fermentation, and only a small amount of glucose is converted to pyruvate for the tricarboxylic acid cycle in the mitochondria [7]. However, when glucose is depleted, yeast cells switch to the respiration mode and begin to use non-fermentable carbon sources, such as ethanol and acetate, to maintain cell activity [8,9].

In the respiration mode, electrons that leak from the electron transport chain can prematurely react with oxygen, resulting in the generation of reactive oxygen species (ROS), such as superoxide anion (O2^•−^), hydrogen peroxide (H_2_O_2_), and hydroxyl radical (^•^OH) [10,11,12]. Glucose starvation enhances mitochondrial ROS production, which can cause redox imbalance and oxidative stress [13]. Other environmental stressors, such as temperature [14,15], osmotic stress [16], and ultraviolet (UV) radiation [17], can also promote mitochondrial oxidative stress in yeast cells. While a low level of ROS is important for cell proliferation and differentiation [18], excess ROS may destroy cellular components and result in cell senescence or death [15]. To protect cells from oxidative damage, enzymatic antioxidant systems within cells, including catalase (CAT), superoxide dismutase (SOD), ascorbate peroxidase (APX), and glutathione peroxidase (GPX), play an important role in the stabilization of intracellular ROS levels [11,15]. Additionally, non-enzymatic antioxidant systems are available in cells, including carotenoids and ascorbic acid and its derivatives, glutathione, proline, and trehalose, among others [19,20].

Carotenoids are important products of red yeasts [21]. Intracellular accumulation of carotenoids is a survival strategy for the cells to adapt to an external environment [21,22]. For instance, to adapt to a cold environment, yeast cells accumulate carotenoids, which alter membrane fluidity at a low temperature [23,24]. When exposed to various stresses, such as osmotic stress, oxidative stress [21,25], or nutrient starvation [26,27,28,29], yeast cells accumulate an increased amount of carotenoids, which confer a protective effect to the cells. During glucose starvation, which is a major nutrient starvation, yeast cells accumulate a higher level of carotenoids [26]. However, the regulatory mechanisms by which glucose starvation promotes carotenoid biosynthesis remain unknown.

The biosynthesis of carotenoids requires a sufficient amount of the substrate acetyl-coenzyme A (acetyl-CoA) [30,31]. The metabolism of acetyl-CoA is complex because it is involved in various metabolic pathways that occur in different cellular compartments [32]. Acetyl-CoA biosynthesis in *S. cerevisiae* occurs mainly through four pathways: (1) in mitochondria, acetyl-CoA is directly formed from pyruvate, a reaction catalyzed by pyruvate dehydrogenase (PDH) complex [33]; (2) in cytoplasm, acetyl-CoA synthetase (ACS) directly catalyzes the conversion of acetate to acetyl-CoA in an ATP-dependent reaction [34]; (3) in peroxisome, acetyl-CoA is the final product of fatty acid β-oxidation [35]; (4) in mitochondria and cytoplasm, transaminases catalyze the conversion of branched-chain amino acids (BCAAs, namely, leucine, isoleucine, and valine) into acetyl-CoA [36,37,38]. However, under glucose starvation, the main metabolic pathways involved in the biosynthesis of acetyl-CoA and carotenoids need to be studied for a deeper understanding. 

*Rhodosporidium* is a well-known producer of carotenoids [39] and lipids [40]. *Rhodosporidium* also has great potential in the production of squalene [41] and sugar alcohol [42,43]. *Rhodosporidium kratochvilovae* YM25235 (isolated from Chenghai Lake, Yunnan, China) is a red yeast that produces polyunsaturated fatty acids (PUFAs) and carotenoids [44,45,46]. Our previous research focused on the adaptability of the YM25235 strain to a low temperature [47,48]. However, in industrial applications, the growth of microorganisms is usually limited by nutrient availability [49]. *S. cerevisiae* responds to the lack of carbon sources [50,51,52,53]. However, past studies have focused mainly on the elucidation of molecular regulation, but not on the changes in downstream metabolites. Moreover, research on the response of *Rhodosporidium* to carbon source limitation is scarce. Therefore, the present study conducted a transcriptomics-based analysis to reveal the adaptive mechanisms of *R. kratochvilovae* YM25235 to glucose starvation.

## 2. Materials and Methods

### 2.1. Yeast Strains and Growth Conditions

The yeast strains used in this study were *R. kratochvilovae* strain YM25235 and the recombinant strain YM25235/pRHRkACOX2. The yeast strains were grown in a yeast extract peptone dextrose (YPD) medium (containing 1% yeast extract, 2% peptone, and 2% glucose) at 28 °C.

### 2.2. Determination of the Consumption of Glucose and Total Nitrogen

The strain YM25235 was fermented for 168 h in 100 mL YPD medium at 160 rpm, 28 °C. Samples of the fermentation broth were checked every 24 h by measuring the optical density (OD) at the 600 nm (cell density) wavelength. After centrifugation (at 4500 rpm for 5 min), 3,5-dinitrosalicylic acid (DNS) [54] undergoes an oxidation–reduction reaction with reducing sugars in the supernatant. The absorbance at a wavelength of 540 nm is measured to detect changes in glucose levels. The regression equation and regression coefficient (R^2^) were calculated by constructing a standard curve of glucose. 

Total nitrogen consumption was detected according to the alkaline potassium persulfate digestion method [55] and using a chemical oxygen demand (COD) multi-parameter photometer (HANNA-hi83399), according to the manufacturer’s instructions for Total Nitrogen High Range Reagents (Hanna Instruments, Smithfield, RI, USA).

### 2.3. RNA Extraction and Sequencing

The strain YM25235 was fermented in 100 mL YPD medium at 160 rpm, 28 °C, and samples were collected in triplicates at 36 h (non-starvation condition) and 96 h (glucose starvation). The samples were centrifuged (at 4500 rpm for 5 min) and washed twice with PBS. Total RNA was extracted from the samples using TRIzol RNA extraction reagent (Thermo Fisher Scientific, Waltham, MA, USA), and the purity and concentration of the extract were determined using the NanoDrop 2000 spectrophotometer (Thermo Fisher Scientific, Waltham, MA, USA), while integrity was assessed using the Bioanalyzer 2100 using the RNA 6000 Nano Kit (Agilent Technologies, Santa Clara, CA, USA). For RNA-seq, the library was prepared using the TruSeq RNA Library Prep Kit v2 (Illumina, San Diego, CA, USA), according to the user manual, and was sequenced using the Illumina NovaSeq6000 platform. The raw reads generated in the fastq format were subjected to an initial quality control test using fastq.

### 2.4. Sequence Read Mapping and Differential Expression Analysis

The original data in the fastq file were preliminarily processed using the fastx_toolkit_0.0.14 [56]. Briefly, clean reads and more reliable results were obtained by removing reads that contained an adapter, low-quality reads with fewer than 50% bases, quality scores lower than 5, and unknown bases that had larger than 10% N bases. Additionally, the GC content, Q20, Q30, and sequence repetition level of the clean reads were calculated. After filtering the clean reads, we used Hierarchical Indexing for Spliced Alignment of Transcripts 2 with standard parameters to map clean reads of high quality to the *R. kratochvilovae* YM25235 genome (NCBI accession number: PRJNA739038), and the quality was evaluated during mapping. Differentially expressed genes (DEGs) were regarded as significant at an arbitrary cut-off value of gene expression fold change (FC) ≥ 2 (absolute value of log_2_ Ratio ≥ 1) and at a q-value of ≤0.001.

### 2.5. Functional Annotation of Gene Transcripts

To further understand which DEGs and pathways were associated with the response of *R. kratochvilovae* YM25235 to glucose starvation, we performed gene ontology (GO) enrichment analysis of all DEGs to identify and categorize the enriched biological process (BP), molecular function (MF), and cellular component (CC). The DEGs were also searched against the Kyoto Encyclopedia of Genes and Genomes (KEGG) [57] database to determine significantly enriched pathways.

### 2.6. Cloning the Full-Length Sequence of the Fatty acyl-CoA oxidase Gene

The cloning experiment was performed as described previously [46]. Briefly, first-strand cDNA of *R. kratochvilovae* YM25235 was synthesized using the HiScript II 1st Strand cDNA Synthesis Kit (Vazyme, Nanjing, China). The full-length coding sequence of the fatty acyl-CoA oxidase gene (*RkACOX2*) was then amplified from the aforementioned cDNA template using Phanta Max Super-Fidelity DNA Polymerase (Vazyme, Nanjing, China), and gene-specific primers were designed according to sequence information obtained from previous RNA-Seq data. The forward primer was 5′-TCACTCACCATGGCGATGCCGACGCCCGTCGATG-3′, and the reverse primer was 5′-CCGGTCGGCATCTACGATATCCTACTCCTTCGCCGGCTT-3′. Subsequently, the amplified *RkACOX2* was subcloned into the pMD18-T vector (Takara, Beijing, China) and then sequenced (Sangon, Shanghai, China). 

### 2.7. Plasmid Construction and Yeast Transformation

The polymerase chain reaction (PCR) amplified *RkACOX2* was subcloned into the pRH2034 vector [58] using the ClonExpress II One Step Cloning Kit (Vazyme, Nanjing, China). The obtained recombinant plasmid pRHRkACOX2 was then validated through restriction enzyme digestion and DNA sequencing analysis. The recombinant plasmid pRHRkACOX2 was further transformed into *R. kratochvilovae* YM25235 using the *Agrobacterium tumefaciens*-mediated transformation method [58,59]. To obtain the recombinant strain YM25235/pRHRkACOX2, positive transformant cells were selected on a YPD agar plate containing hygromycin B (150 µg/mL) and confirmed by PCR assay.

### 2.8. Lipid Extraction and Separation

Yeast strains were cultured in YPD medium for 168 h, and the cells were collected, washed with PBS, and freeze-dried. Total lipids were extracted according to the Bligh and Dyer method [60]. Briefly, 1.6 mL of hydrochloric acid (4 mol/L) was added to 0.4 g of dry cells, after which they were mixed thoroughly and left at room temperature for 30 min; the mixture solution was placed in a boiling water bath for 3 min and then cooled to room temperature. Then, 3.2 mL of chloroform/methanol solution (2:1, *v*/*v*) was added to the mixture solution, after which the solution was mixed and centrifuged at 8000 rpm for 5 min. The organic phase was collected into a clean tube, and lipid extraction was repeated twice. The extracts were dried, weighed, dissolved in chloroform, and stored at −20 °C.

The extracted total lipids were separated into triacylglycerols (TAGs) and free fatty acids (FFAs) by thin-layer chromatography (TLC) [61,62], for which silica gel plates GF254 of 20 × 20 cm were used. Plates with spots containing 10 mg of sample were separated using a solvent mixture consisting of petroleum ether/ethyl ether/acetic acid (70:30:1, *v*/*v*/*v*). Lipid spots were screened by exposing the plate to a chromogenic agent (anisaldehyde/acetic acid/concentrated sulfuric acid, 1:100:2, *v*/*v*/*v*) under a UV lamp (λ = 254 nm). TAGs and FFAs were identified by comparing the samples with the standards applied in the TLC plates. Then, each identified spot was scraped and solubilized with chloroform.

### 2.9. Fatty Acid Analysis

Fatty acid profiling of TAGs and FFAs was performed by analyzing the presence of fatty acid methyl esters (FAMEs) [44,63]. Briefly, fatty acids were extracted by adding 2.5 mL of 10% KOH onto TAGs or FFAs (3 mg), followed by saponification at 70 °C for 3–5 h. The pH of the product was adjusted to 2.0 with hydrochloric acid, and the extracted fatty acids were methyl-esterified in 4 mL of 14% boron trifluoride in methanol at 70 °C for 1.5 h. Then, FAME was extracted with hexane. FAME was analyzed using gas chromatography–mass spectrometry (GC-MS) [64]. The instrument and column used were the GCMS-QP2010 SE (Shimadzu, Kyoto, Japan) and HP-88 (100 m × 0.25 mm × 0.20 μm; Agilent). 

### 2.10. Acetyl-CoA Analysis

The strains YM25235 and YM25235/pRHRkACOX2 were fermented in 100 mL YPD medium for 120 h at 160 rpm, 28 °C. Samples were collected for measuring acetyl-CoA content using a detection kit (Solarbio, Beijing, China).

### 2.11. Total Carotenoids and Components Analysis

For cell disruption and carotenoid extraction [65], 400 mg dry cells was disrupted by grinding them with SiO_2_, and carotenoids were extracted with 10 mL of acetone at room temperature. The extract solution was centrifuged at 4500 rpm for 5 min, and the supernatant was collected, while the precipitate was ground and subjected to a second extraction. The supernatants from the two extractions were mixed together, and the OD was measured at the 445 nm (β-carotene) wavelength to determine the total carotenoid content [66]. All procedures were carried out in triplicate.

The carotenoid composition was determined using high-performance liquid chromatography (HPLC) [67]. The instrument and column used were the Agilent 1260 Infinity II and ZORBAX Eclipse XDB-C18 (4.6 mm × 250 mm × 5 µm; Agilent). Torulene, torularhodin, γ-carotene, and β-carotene pigments (CaroteNature, Münsingen, Switzerland) were used as standards for the HPLC quantitative analysis.

### 2.12. Determination of Total Intracellular ROS

Yeast strains were cultured continuously in a YPD medium and sampled every 24 h; intracellular ROS levels were measured using 2′,7′-dichlorodihydrofluorescein diacetate (DCFH-DA) (Beyotime, Shanghai, China). Briefly, cells (5 × 10^6^) were collected and washed twice with 0.05 M PBS (pH 7.0). Then, DCFH-DA was added at a final concentration of 10 µM, and the cells were incubated in the dark for 30 min at 30 °C. Cells were washed twice with 0.05 M PBS (pH 7.0) and resuspended in 1 mL of 0.1 M PBS (pH 7.0). Data were measured using a microplate reader (Infinite M200 PRO; Tecan, Männedorf, Switzerland) at excitation/emission wavelengths of 488/525 nm.

### 2.13. Quantitative Real-Time PCR (qPCR) Validation

To confirm the RNA-seq results, 15 representative genes (chosen from up-regulated and down-regulated DEGs) were selected for qPCR analysis, and experiments were performed with three biological replicates. The primers used in the qPCR assay were designed automatically in NCBI (Appendix A). Total RNA was reverse-transcribed into cDNA using the HiScript II 1st Strand cDNA Synthesis Kit (Vazyme, Nanjing, China). The qPCR assay was performed on the Bio-Rad CFX96 Fast Real-Time PCR System (Bio-Rad, Hercules, CA, USA) with the ChamQ SYBR qPCR Master Mix (Vazyme, Nanjing, China). The small subunit rRNA (SSU rRNA) was used as the reference gene. The qPCR assay was repeated three times for each gene in each sample, and experiments were performed with three biological replicates. The 2^−ΔΔCT^ method was used to normalize the expression results of each gene, and the final results were presented as mean ± standard deviation [68]. 

The effects of *RkACOX2* overexpression at the transcription level of genes related to fatty acid metabolism, carotenoid biosynthesis, and antioxidants under conditions of glucose starvation were also validated using the qPCR assay, and the processes were the same as described above.

## 3. Results

### 3.1. Effects of Glucose Consumption on Cell Growth, Total ROS Levels, and Carotenoid Biosynthesis

Glucose starvation affected carotenoid biosynthesis by *R. kratochvilovae* YM25235 (Figure 1). When YM25235 was cultured in the YPD medium, the glucose present in the medium was fully consumed and could barely be detected after 72 h, resulting in glucose starvation and the arrest of cell growth. However, the nitrogen content in the medium remained 50% until 192 h (Appendix A), which indicated the adequate availability of nitrogen. After glucose depletion in the medium (at 72 h), the total ROS levels and carotenoid content in the cells increased rapidly. With the accumulation of carotenoids, the total ROS levels showed a decreasing trend after 120 h. To investigate the relationship between glucose starvation and carotenoid accumulation, we used the 36 h samples (non-starvation condition) as the control group and the 96 h samples (glucose starvation) as the treatment group for transcriptome sequencing. 

### 3.2. Transcriptome Sequencing and Assembly

Six transcriptome libraries from the non-starvation condition (at 36 h) and glucose starvation condition (at 96 h) were constructed. The RNA-seq data have been submitted to the NCBI under accession number PRJNA937027. A total of 45,289,004, 49,222,456, 47,656,900, 45,888,648, 43,176,246, and 47,351,146 clean reads for the triplicate of the non-starvation condition (at 36 h) and glucose starvation condition (at 96 h) were obtained, after which they were filtered by removing redundant transcripts (Table 1). High-quality clean reads were mapped to the genome of *R. kratochvilovae* YM25235, and a high percentage of the clean reads were uniquely mapped to the selected genome. 

### 3.3. DEGs under Glucose Starvation Conditions

All DEGs were identified and compared between the treatment and control groups. Overall, 1661 and 995 genes had significantly up-regulated and significantly down-regulated expression, respectively (Appendix A). The top 10 genes that had the most significant changes in expression under the glucose starvation condition were further analyzed (Table 2). The DEGs with the most significantly up-regulated expression were related to antioxidation and biological regulation; for example, OsmC family protein (*OsmC*) with antioxidant function, transcription factor Hsf1 (*Hsf1*) involved in protein folding, serine/threonine-protein phosphatase 2A activator 2 (*PP2A*) involved in the regulation of signaling pathways and physiological processes, and cell division control protein (*CDC*) involved in cell division and cell cycle regulation. The top three DEGs with the most significantly down-regulated expression were ABC drug exporter AtrF (*AtrF*), zip-like iron-zinc transporter (*ZIP*), and putative phospholipid-binding protein (*Cts1*). A few DEGs were hypothetical proteins, that is, their functions could not be annotated.

### 3.4. Verification of RNA-Sequencing Data

The qPCR assay was performed to validate the results of RNA-seq analysis. Expression of the selected DEGs obtained from qPCR analysis was in agreement with the RNA-seq results (Figure 2). The RNA-seq and qPCR data suggested a significantly positive correlation (Pearson’s correlation coefficient = 0.8286, *p* < 0.001), according to linear regression analysis (Figure 2), confirming the reliability and reproducibility of the RNA-seq data.

### 3.5. Functional Enrichment of the DEGs

In a pairwise comparison of the treatment group versus control group, DEGs were significantly enriched as follows (*p*-value < 0.01): 21 terms for BP, 14 terms for CC, and 9 terms for MF (Figure 3). For BP terms, the three GO terms with the greatest number of DEGs were metabolic process, cellular process, and localization. Moreover, some BP terms involving the starvation response were detected, such as response to stimulus, regulation of BP, and signaling. For CC terms, the top three terms were cell, cell membrane, and cell organelle. For MF terms, the top three terms were catalytic activity, catalytic binding, and transporter activity. In short, the DEGs with up-regulated and down-regulated expression in the treatment group were classified into various functions, including transcriptional regulation, signal transmission, protein binding, and response to a stimulus.

The DEGs with up-regulated and down-regulated expression were related to 119 different KEGG pathways (Appendix A), in which 22 pathways were significantly enriched (*p*-value < 0.05) (Figure 4). These 22 pathways are related to acetyl-CoA biosynthesis, antioxidant mechanism, signal transduction, and regulation of biological processes. The pathways related to acetyl-CoA biosynthesis are fatty acid degradation, glyoxylate and dicarboxylate metabolism, pyruvate metabolism, ascorbate and aldarate metabolism, inositol phosphate metabolism, and metabolism of various amino acids (valine, leucine, isoleucine, tryptophan, arginine, proline, and lysine). These pathways can provide raw material and energy for other biosynthetic pathways. The antioxidant-related pathway (peroxisome) participates in defense against oxidative stress [11,20]. Most of the DEGs were associated with the mitogen-activated protein kinase (MAPK) pathway. MAPK is an important pathway involved in transmitting signals from the cell surface to the nucleus [69]. Moreover, cell cycle, transcription, cellular transport, growth/morphogenesis, and signal transduction pathways were enriched in KEGG under the glucose starvation condition. These biological processes have a positive effect on inducing a response to glucose starvation.

### 3.6. Effect of RkACOX2 Overexpression on FFA Levels under Glucose Starvation

Fatty acids are important products of YM25235. To further understand the changes in fatty acid levels under glucose starvation, the first key rate-limiting enzyme gene (*RkACOX2* from *R. kratochvilovae* YM25235) (GenBank Access ID: OQ543117) of the peroxisomal β-oxidation system was overexpressed. The recombinant strain obtained was YM25235/pRHRkACOX2. Although the amount of total lipids in the YM25235/pRHRkACOX2 strain was not significantly different from that of the control YM25235 strain (Appendix A), significant changes in FFA levels were found in the YM25235/pRHRkACOX2 strain (Table 3). The YM25235/pRHRkACOX2 strain had a significantly lower level of palmitic acid (C16:0), linoleic acid (LA, C18:2), and α-linolenic acid (ALA, C18:3) than the YM25235 strain (decreased by 9.22%, 37.59%, and 32.39%, respectively), while palmitoleic acid (16:1) levels were increased. The results showed that overexpression of *RkACOX2* mainly led to the degradation of LA and ALA in FFAs and the redistribution of FFA categories.

### 3.7. Effect of RkACOX2 Overexpression on Intracellular acetyl-CoA Levels under Glucose Starvation

To determine the changes in intracellular acetyl-CoA levels after *RkACOX2* overexpression, we analyzed the acetyl-CoA levels in YM25235 and YM25235/pRHRkACOX2 strains. The acetyl-CoA level in YM25235/pRHRkACOX2 (465.32 nmol/g) was 43.44% higher than that in the control strain (324.39 nmol/g) (Figure 5). This indicates that, under glucose starvation, overexpression of the *RkACOX2* gene increases acetyl-CoA biosynthesis in cells.

### 3.8. Effect of RkACOX2 Overexpression on Carotenoid Biosynthesis under Glucose Starvation

Carotenoids are important products of *R. kratochvilovae* YM25235. To investigate the effect of *RkACOX2* overexpression on carotenoid biosynthesis under the glucose starvation condition, carotenoids were quantified in this study. YM25235/pRHRkACOX2 produced more total carotenoids than the wild-type under glucose starvation (7.59 mg/g DCW and 5.68 mg/g DCW, respectively; Table 4). HPLC analysis further revealed that β-carotene was the predominant component of the two strains. The levels of both torulene and β-carotene were significantly increased in YM25235/pRHRkACOX2 compared with those in YM25235 (100% and 19.87% increase, respectively). These results suggest that *RkACOX2* overexpression had a positive effect on carotenoid biosynthesis under glucose starvation.

### 3.9. Effect of RkACOX2 Overexpression on the Transcription of Genes Related to Fatty Acid Oxidation, Carotenoid Biosynthesis, and Antioxidants under Glucose Starvation

qPCR results confirmed that *RkACOX2* overexpression significantly promoted the expression of the ACYL-COA OXIDASE gene (*RkACOX2*), key genes in the mevalonate (MVA) pathway (*RkAcaT2*, *RkMK*, *RkHMGCS*, and *RkHMGCR*), genes related to carotenoid biosynthesis (*RkCrtYB* and *RkCrtI*), and antioxidant-related genes (*RkCTT1* and *RkSOD2*) at the transcription level under glucose starvation (Figure 6). However, in the presence of glucose, the expression levels of genes related to carotenoid synthesis and antioxidants were significantly reduced (Appendix A). Thus, fatty acid oxidation not only increased acyl-CoA content, but also enhanced resistance to oxidation in YM25235 after glucose was depleted.

## 4. Discussion

It is generally observed that most abiotic stresses cause ROS accumulation in cells. One of our previous studies also confirmed a negative correlation between the increase in carotenoid biosynthesis and ROS accumulation at low temperatures [48]. The lack of a carbon source inevitably induces ROS production [13,26], a phenomenon similar to that encountered in this study. When glucose is depleted, the level of cellular ROS increases (Figure 1). To maintain ROS at low steady-state levels, the carotenoid content has to be increased, and a decrease in intracellular ROS levels can be clearly detected after 120 h (Figure 1). Recently, our analysis similarly showed that inhibition of carotenoid biosynthesis through the knockout of the key gene *RkCrtYB* in the carotenoid biosynthesis pathway in *R. kratochvilovae* YM25235 under glucose starvation caused an increase in intracellular ROS (Appendix A). However, in the early stages of cell proliferation and growth, cells may generate ROS due to protein folding and disulfide bond formation [70], while synthesis of carotenoids in *R. kratocvilovae* YM25235 has not yet significantly occurred (Figure 1b). Thus, during early stages, the inadequate antioxidant capacity of *R. kratocvilovae* may lead to the accumulation of ROS byproducts within the cells. Additionally, *CAT* is highly induced in *R. kratochvilovae* YM25235 under glucose starvation (Appendix A). H_2_O_2_ may be the major intracellular ROS induced by glucose starvation in YM25235 because CAT protects cells from H_2_O_2_ toxicity by breaking it down into O_2_ and H_2_O [71]. Overall, both enzymatic and non-enzymatic antioxidant systems of *R. kratochvilovae* YM25235 were activated to resist glucose starvation.

The substrate for carotenoid biosynthesis in yeast cells is acetyl-CoA [31]. In industrial production, the development of different strategies to improve the acetyl-CoA pool is an effective way to promote the downstream production of carotenoid [72]. Based on the transcriptome analysis of *R. kratochvilovae* YM25235, fatty acid β-oxidation, amino acid metabolism, and pyruvate metabolism are the main metabolic pathways that can increase the acetyl-CoA pool under glucose starvation (Figure 4). In KEGG (Figure 4), the fatty acid degradation pathway was significantly enriched, and the enzymes involved, such as long-chain acyl-CoA synthetase (*LACS*), acyl-coenzyme A oxidase 2 (*ACOX2*), and 3-ketoacyl-CoA thiolase (*ACAA*), were all significantly induced at the transcription level (Appendix A). Additionally, the contribution of fatty acid degradation to the acetyl-CoA pool was confirmed by overexpression of *RkACOX2*, the gene of the first key rate-limiting enzyme of the peroxisomal β-oxidation system. *RkACOX2* overexpression promoted FFAs degradation and carotenoid production in *R. kratochvilovae* YM25235 (Table 3 and Table 4, Figure 5 and Figure 6). Our results were in line with a study [73] that *ACOX2* knockout enhances lipid productivity on *Rhodotorula*. However, in the presence of glucose, *ACOX2* overexpression did not result in an increase in carotenoid production (Appendix A). A positive correlation existed between fatty acid degradation and carotenoid biosynthesis, which is consistent with other research [30,74]. This indicates that peroxisomal fatty acid β-oxidation is an effective way to improve the acetyl-CoA level and subsequently promote carotenoid production when yeast cells respond to environmental stress. Overall, this study has preliminarily established the correlation between fatty acid degradation and carotenoid biosynthesis under glucose starvation, but the exact molecular mechanisms need to be further confirmed through gene knockout studies or other approaches. 

Cellular amino acid metabolic pathways are also regulated under carbon starvation [75,76] and undergo changes in the acetyl-CoA pool. In this study, the metabolic pathways of valine, leucine, isoleucine, tryptophan, arginine, proline, and lysine were enriched in *R. kratochvilovae* YM25235 under glucose starvation (Figure 4). These amino acids may be used for producing precursors of important molecules involved in alternative carbon metabolism in *R. kratochvilovae* YM25235. In particular, BCAAs (leucine, isoleucine, and valine) showed significant degradation under glucose starvation. At the transcription level, BCAA degradation-related enzymes were highly up-regulated, including branched-chain amino acid aminotransferase (*BCAT*), branched-chain α-ketoacid dehydrogenase complex (BCKDHA and BCKDHB), and 2-oxoacid dehydrogenase acyltransferase (*DBT*) (Appendix A). During catabolism, BCAAs can be converted into acetyl-CoA or succinyl-CoA [37] and can then provide additional substrates for other metabolisms. Some amino acids are metabolic regulators in various metabolic pathways [77]. For example, tryptophan is the precursor of nicotinic acid (vitamin B3), 5-hydroxytryptamine, melatonin, NAD, and NADP [78], among others; these metabolites have an important impact on the physiological and biochemical reactions of cells.

During glucose starvation, acetyl-CoA can be synthesized through PDH bypass by yeast cells [34,79]. The transcriptome analysis of this study showed that, under glucose starvation, the key genes in the PDH bypass, namely, the acetyl-CoA synthase (ACS) gene and aldehyde dehydrogenase (ALDH) gene, were significantly enriched (Appendix A). As expected, overexpression of the *ACS1*, *ACS2*, and *ALDH* genes in *R. kratochvilovae* YM25235 significantly improved intracellular acetyl-CoA levels (Appendix A). The design of the PDH bypass is also an effective strategy to improve the acetyl-CoA level in the cytoplasm and carotenoid production [72].

## 5. Conclusions

In conclusion, a higher level of ROS was observed in the YM25235 strain under glucose starvation. To adapt to glucose starvation, the yeast cells had increased production of carotenoids and antioxidant enzymes to decrease ROS levels. As shown in the transcriptome analysis results, the pathways involved in the biosynthesis of acetyl-CoA were up-regulated, including fatty acid β-oxidation, amino acid metabolism, and pyruvate metabolism. These pathways provided a carbon source and energy for various biological reactions and cell maintenance. In YM25235, which is an oleaginous yeast strain, fatty acid degradation contributed to the acetyl-CoA pool and carotenoid biosynthesis, which was confirmed by *RkACOX2* gene overexpression. This study provides comprehensive information on the adaptive mechanisms of *R. kratochvilovae* YM25235 under glucose starvation at the transcriptional level, which may provide a reference for the large-scale production of carotenoids using *Rhodosporidium*. 

## Figures and Tables

**Figure 1 microorganisms-11-02168-f001:**
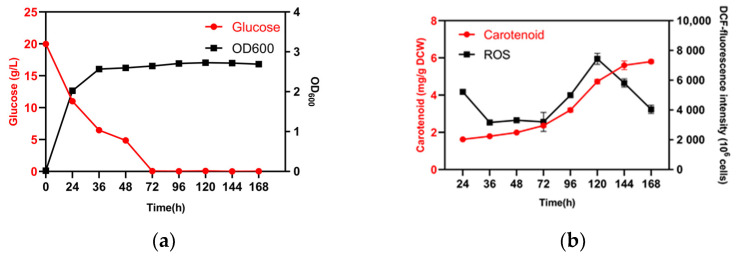
Effects of glucose consumption on cell growth, total ROS levels, and carotenoid biosynthesis. (**a**) Glucose consumption and cell growth. Glucose (red): g/L; OD_600_ (black). (**b**) Total ROS levels and carotenoid biosynthesis. Carotenoid (red): mg/g DCW; ROS (black): DCF-fluorescence intensity (10^6^ cells). DCW: dry cell weight; OD: optical density; ROS: reactive oxygen species.

**Figure 2 microorganisms-11-02168-f002:**
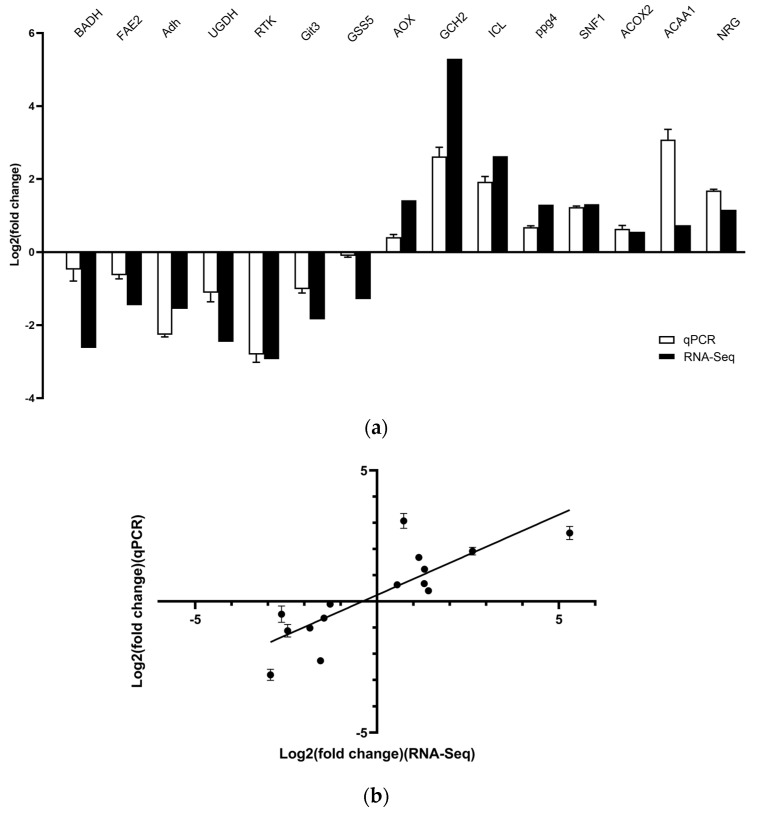
Confirmation of RNA-seq results by qPCR assay. (**a**) Comparison of the fold change in the expression level of 15 selected DEGs as detected using RNA-seq and qPCR. (**b**) Correlation of the fold change in the expression of 15 DEGs analyzed by RNA-seq and qPCR. DEG: differentially expressed gene; qPCR: quantitative polymerase chain reaction.

**Figure 3 microorganisms-11-02168-f003:**
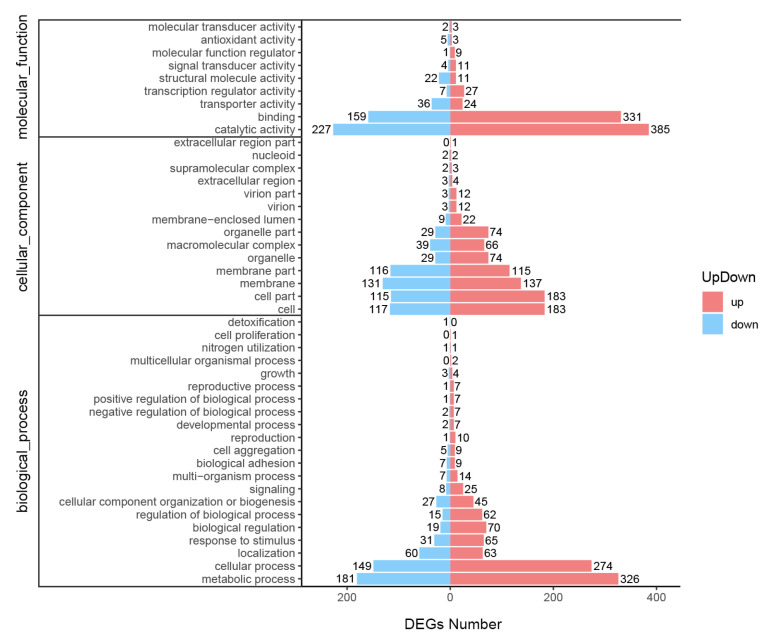
Vertical histogram of DEG-enriched GO terms. BP: biological process; CC: cellular component; MF: molecular function; DEG: differentially expressed gene; GO: gene ontology.

**Figure 4 microorganisms-11-02168-f004:**
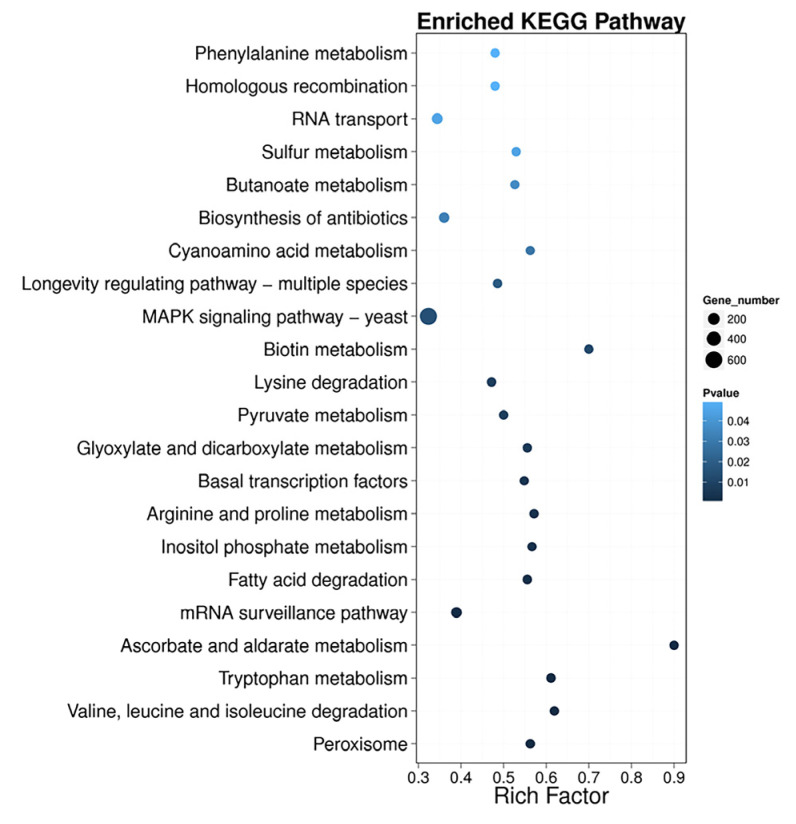
Top 22 differential gene pathway enrichment results. The X-axis represents the enrichment factor value, and the Y-axis represents the pathway name. Color represents the *p*-value, with smaller values representing more significant enrichment results. The size of the dots represents the number of DEGs. Rich Factor refers to the enrichment factor value, and a larger value indicates a more pronounced enrichment result. Rich Factor was calculated as the number of differentially expressed genes under the pathway term divided by the total number of annotated genes under that pathway term.

**Figure 5 microorganisms-11-02168-f005:**
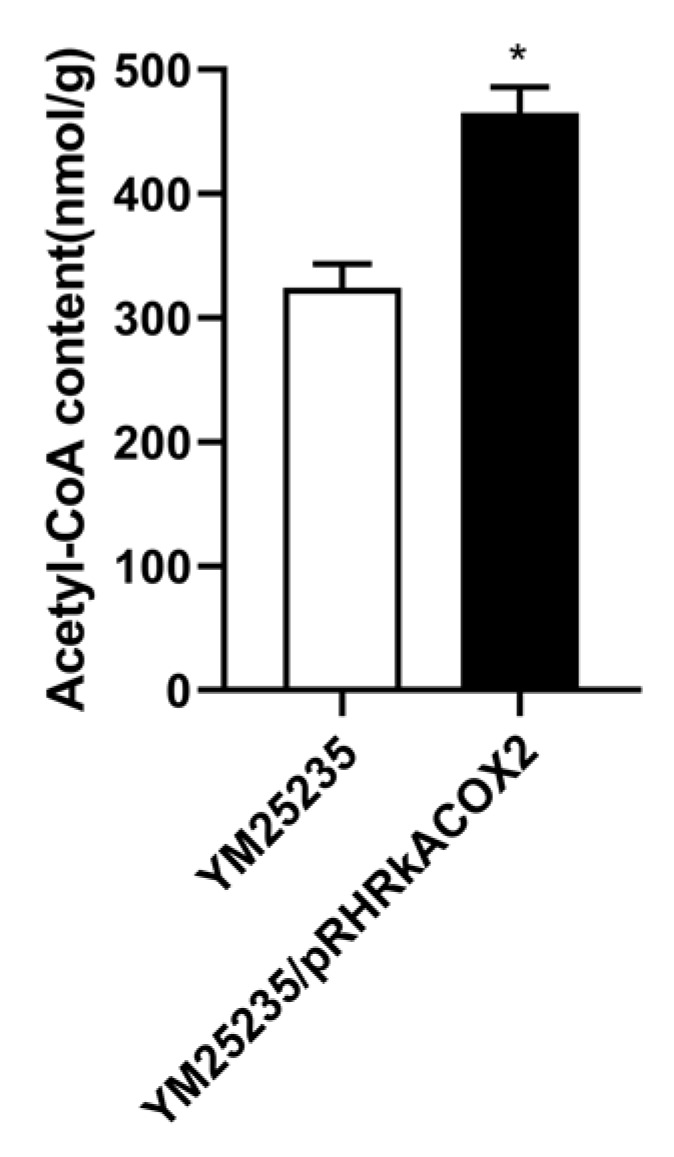
Effect of *RkACOX2* overexpression on the acetyl-CoA level under glucose starvation. Graphs show mean ± standard deviation of three independent experiments. Statistically significant differences are indicated (* *p* < 0.05).

**Figure 6 microorganisms-11-02168-f006:**
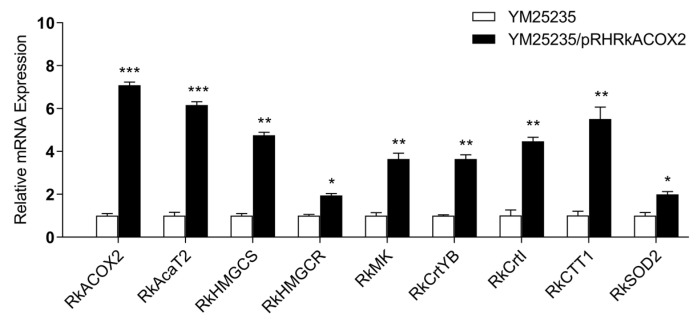
Effect of *RkACOX2* overexpression on the transcription level of genes related to fatty acid oxidation, carotenoid biosynthesis, and antioxidants under glucose starvation. Graphs are presented as mean ± standard deviation of three independent experiments. Statistically significant differences are indicated (* *p* < 0.05, ** *p* < 0.01, *** *p* < 0.001). *RkACOX2*: fatty acyl-CoA oxidase; *RkAcaT*: acetoacetyl-CoA thiolase; *RkHMGCS*: HMG-CoA synthase; *RkHMGCR*: HMG-CoA reductase; *RkMK*: mevalonate kinase; *RkCrtYB*: phytoene synthase/lycopene cyclase; *RkCrtI*: phytoene dehydrogenase; *RkCTT1*: catalase; *RkSOD2*: superoxide dismutase.

**Table 1 microorganisms-11-02168-t001:** Statistical analysis of the quality of clean reads and mapping of genes.

Sample	Raw Reads	Clean Reads	Clean Bases	Total Reads	Total Mapped
36_1	45,807,814	45,289,004	6.75 G	45,289,004	40,325,735 (89.04%)
36_2	49,754,936	49,222,456	7.31 G	49,222,456	43,862,890 (89.11%)
36_3	48,145,136	47,656,900	7.09 G	47,656,900	42,238,415 (88.63%)
96_1	47,234,546	45,888,648	6.58 G	45,888,648	40,276,865 (87.77%)
96_2	43,789,462	43,176,246	6.42 G	43,176,246	37,087,034 (85.90%)
96_3	48,064,754	47,351,146	7.05 G	47,351,146	40,812,496 (86.19%)

**Table 2 microorganisms-11-02168-t002:** Top 10 differentially expressed genes with the most up-regulated and down-regulated expression.

Gene id	log2 Ratio (Treatment/Control)	q-Value	Gene Description	Gene Symbol
The top 10 most up-regulated DEGs				
EVM0002799	12.16840050	2.64 × 10^−212^	OsmC family protein	*OsmC*
EVM0007334	12.06697468	1.04 × 10^−201^	Transcription factor Hsf1	*Hsf1*
EVM0000758	12.01547883	1.57× 10^−196^	hypothetical protein	\
EVM0000848	11.74144109	4.95 × 10^−171^	regulator of gluconeogenesis	*RGG*
EVM0000135	11.72418730	1.55 × 10^−169^	leucine-rich repeat domain-containing protein	*LRR*
EVM0001989	11.60200485	2.60 × 10^−159^	hypothetical protein	\
EVM0006562	10.57452947	7.47 × 10^−94^	Serine/threonine-protein phosphatase 2A activator 2	*PP2A*
EVM0008373	10.56314106	2.98 × 10^−89^	alpha-ketoglutarate dependent 2,4-dichlorophenoxyacetate dioxygenase	*TfdA*
EVM0008165	10.07478880	2.74 × 10^−72^	cleavage and polyadenylation specific factor 5	*Cpsf5*
EVM0000497	9.601556094	2.38 × 10^−56^	cell division control protein	*CDC*
The top 10 most down-regulated DEGs				
EVM0004206	−11.90660765	3.86 × 10^−308^	ABC drug exporter AtrF	*AtrF*
EVM0002079	−11.27993663	2.37 × 10^−223^	zip-like iron-zinc transporter	*ZIP*
EVM0005177	−11.05445575	1.06 × 10^−198^	putative Phospholipid binding protein	*Cts1*
EVM0000550	−10.80320920	2.71 × 10^−174^	acyl-CoA thioester hydrolase	*ACOT*
EVM0002631	−10.65221864	4.94 × 10^−161^	ribosomal protein L9 family protein	*RPL9*
EVM0004367	−10.64333877	2.77 × 10^−160^	glutamate-rich WD repeat-containing protein	*GRWD*
EVM0001224	−10.60837800	5.05 × 10^−144^	3-carboxymuconate cyclase	*CMC*
EVM0002253	−10.44011035	1.59 × 10^−143^	hypothetical protein	\
EVM0006484	−10.38925953	1.06 × 10^−139^	zinc finger, RING-type protein	*RZF*
EVM0003900	–10.38419536	2.24 × 10^−111^	macrofage activating glycoprotein	*GcMAF*

**Table 3 microorganisms-11-02168-t003:** Analysis of fatty acid composition under glucose starvation.

Fatty Acid (mg/g DCW)	FFAs		TAGs	
	YM25235	YM25235/pRHRkACOX2	YM25235	YM25235/pRHRkACOX2
C16:0	3.47 ± 0.08	3.15 ± 0.03 *	2.43 ± 0.02	3.05 ± 0.01 ^&&^
C16:1	0.69 ± 0.06	0.98 ± 0.03 *	0.42 ± 0.02	0.64 ± 0.08
C18:0	0.57 ± 0.01	0.53 ± 0.04	0.65 ± 0.07	0.73 ± 0.02
C18:1OA	13.22 ± 0.08	13.47 ± 0.33	6.47 ± 0.30	8.48 ± 0.35 ^&^
C18:2LA	5.40 ± 0.08	3.37 ± 0.19 **	2.13 ± 0.16	1.88 ± 0.26
C18:3ALA	0.71 ± 0.02	0.48 ± 0.03 *	0.22 ± 0.06	0.25 ± 0.03

Data are presented as mean ± standard deviation of three independent experiments. Statistically significant differences are indicated (* *p* < 0.05; ** *p* < 0.01, compared with FFA; ^&^ *p* < 0.05; ^&&^ *p* < 0.01, compared with TAG). FFAs: free fatty acids; TAGs: triacylglycerols.

**Table 4 microorganisms-11-02168-t004:** Analysis of carotenoid composition under glucose starvation.

Strain	Total Carotenoid (mg/g DCW)	Carotenoid Composition (mg/g DCW)			
		Torularhodin	Torulene	γ-carotene	β-carotene
YM25235	5.68 ± 0.23	1.65 ± 0.29	0.46 ± 0.09	0.40 ± 0.29	3.17 ± 0.13
YM25235/pRHRkACOX2	7.59 ± 0.35 *	1.52 ± 0.11	0.92 ± 0.11 *	1.35 ± 0.34	3.80 ± 0.04 *

Data are presented as mean ± standard deviation of three independent experiments. Statistically significant differences are indicated (* *p* < 0.05).

## Data Availability

All raw read data generated in this study have been deposited on the National Center for Biotechnology Information platform (NCBI) under the BioProject ID: PRJNA937027. The coding sequence of *RkACOX2* was deposited in GenBank under the accession number OQ543117.

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
