# Peer review of "Comprehensive Response of Rhodosporidium kratochvilovae to Glucose Starvation: A Transcriptomics-Based Analysis"

_microorganisms, 2023, doi:10.3390/microorganisms11092168_

Round 1
Reviewer 1 Report
In this paper, the authors performed transcriptomics-based analyses in the yeast Rhodosporidium kratochvilovae under the glucose starvation condition. The paper reports that genes involved in the β-oxidation process, including ACOX2, are upregu lated in the starvation condition. Further genetic analyses show a link between the fatty acid degradation and carotenoid generation processes. The results are likely useful further studies and applications of this yeast. My comments are followings:
1) Line 415-416, “the correlation between fatty acid degradation and carotenoid biosynthesis under glucose starvation has been well established in this study”: The authors merely performed genetic/transcriptomic analysis to show the link, so I am not sure if the evidences are sufficient to describe that their correlation is “well established”. It may be overstatement.
2) Do you have any data or idea that the link between fatty acid degradation and carotenoid biosynthesis is also seen in the presence of glucose (i.e. are carotenoid synthesis genes induced in ACOX2-overexpressing cells under the normal condition)? Such information would be useful to use R. kratochvilovae for production of substances.
Minor point:
1) Line 322-323, “The fatty acid is an important product of YM25235 to understand the changes in fatty 322 acid levels after glucose depletion”: This sentence is not clear.
2) There are several typo mistakes.
For example:
Line 188: 28 ° C
Line 206: 5 x 106
Reviewer 3 Report

as in 1 comments
Round 2
Reviewer 2 Report
I appreciate the consideration of my comments by the authors. They have made sufficient changes to the manuscript to merit its publication in Microorganisms.
Reviewer 3 Report
I have difficulty to apreciate final manuscript presentation, because of simultaneous presence of corrections.